# Upper Airway-Related Symptoms According to Mental Illness and Sleep Disorders among Workers Employed by a Large Non-Profit Organization in the Mountain West Region of the United States

**DOI:** 10.3390/ijerph20247173

**Published:** 2023-12-13

**Authors:** Ray M. Merrill, Ian S. Gibbons, Christian J. Barker

**Affiliations:** Department of Public Health, College of Life Sciences, Brigham Young University, Provo, UT 84602, USA; gibbonsian6@gmail.com (I.S.G.); cjb18@student.byu.edu (C.J.B.)

**Keywords:** confounder, medical claim, mental illness, rates, sleep disorder

## Abstract

The relationships between selected upper airway-related symptoms (speech disturbances, voice disorders, cough, and breathing abnormalities) and mental illness and sleep disorders have been previously demonstrated. However, these relationships have not been compared in a single study with consideration of potential confounding variables. The current research incorporates a descriptive study design of medical claims data for employees (~21,362 per year 2017–2021) with corporate insurance to evaluate the strength of these relationships, adjusting for demographic variables and other important confounders. The upper airway-related symptoms are each significantly and positively associated with several mental illnesses and sleep disorders, after adjusting for demographic and other potential confounders. The rate of any mental illness is 138% (95% CI 93–195%) higher for speech disturbances, 55% (95% CI 28–88%) higher for voice disorders, 28% (95% CI 22–34%) higher for cough, and 58% (95% CI 50–66%) higher for breathing abnormalities, after adjustment for the confounding variables. Confounding had significant effects on the rate ratios involving cough and breathing abnormalities. The rate of any sleep disorder is 78% (95% CI 34–136%) higher for speech disturbances, 52% (95% CI 21–89%) higher for voice disorders, 34% (95% CI 27–41%) higher for cough, and 172% (95% CI 161–184%) higher for breathing abnormalities, after adjustment for the confounding variables. Confounding had significant effects on each of the upper airway-related symptoms. Rates of mental illness and sleep disorders are positively associated with the number of claims filed for each of the upper airway-related symptoms. The comorbid nature of these conditions should guide clinicians in providing more effective treatment plans that ultimately yield the best outcome for patients.

## 1. Introduction

Speech disturbances, voice disorders, cough, and breathing abnormalities have comorbid relationships with mental illness and sleep disorders. For example, speech disturbances are comorbid with anxiety [1]; voice disorders are comorbid with anxiety, major depressive episodes, phobia, and overall mental health problems [2,3]; chronic cough is comorbid with severe mental health disturbances [4]; and breathing abnormalities are comorbid with stress, anxiety, depression, and panic disorders [5,6]. Further, poor vocal quality is related to poor sleep quality [7,8], and chronic cough and breathing abnormalities are related to sleep apnea [9,10,11,12].

Upper airway-related symptoms like speech disturbances, voice disorders, cough, and breathing abnormalities are often studied together. For example, research has related each of them and their combination to quality-of-life measures [13,14]. We also see their complex comorbid relationships with mental illnesses and sleep disorders [1,2,3,4,5,6,7,8,9,10,11,12]. However, it is not known whether their relationships are similar and remain statistically significant after adjusting for certain potential confounders. If an underlying risk factor confuses some or all of the relationships, that needs to be considered in determining appropriate treatment.

A potential confounder is a common risk factor for both the exposure and outcome variables. Controlling for potential confounders is important for establishing the existence and strength of association between upper airway-related symptoms and mental illness and sleep disorders. In the current study, we consider five potential confounders (i.e., gastroesophageal reflux, asthma, allergies, sinusitis, and hypertension) of the associations between upper airway-related symptoms and mental illness and sleep disorders. These potential confounders are considered because of their association with both upper airway-related symptoms and mental illness and sleep disorders. The studies showing relationships between these variables [1,2,3,4,5,6,7,8,9,10,11,12] did not adjust for these potential confounders, except in the study showing an association between chronic cough and mental health disturbances [4], where they adjusted for angiotensive-converting enzyme (ACE) inhibitors (treatment for hypertension). Previous research has shown that speech disturbances, voice disorders, cough, and breathing abnormalities are associated with gastroesophageal reflux [15,16,17], asthma [18,19,20,21], allergies [22,23,24,25], sinusitis [26,27,28], and hypertension [29,30,31]. In addition, gastroesophageal reflux is associated with anxiety and depression [32]; asthma is associated with attention-deficit/hyperactivity disorder (ADHD), anxiety, and major depression [33,34]; allergies are associated with psychiatric disorders [35,36]; sinusitis is associated with depression [37]; hypertension is associated with anxiety, depression, impulsive eating disorders, and substance use disorders [38]; and gastroesophageal reflux, asthma, allergies, sinusitis, and hypertension are each associated with sleep disorders [39,40,41,42,43].

The purpose of the current descriptive study is to determine and compare the strength of associations between speech disturbances, voice disorders, cough, and breathing abnormalities and selected mental illnesses and sleep disorders while statistically adjusting for demographic and five important variables. The potential confounders will be shown to correlate with both the upper airways-related symptoms and, independent of those relationships, correlate with the mental illness and sleep disorder variables. The study is based on employee healthcare claims data from a large non-profit organization in the Mountain West Region of the United States. While other studies have examined some of the relationships considered in this study, our comprehensive assessment of multiple upper airway-related symptoms and mental illness and sleep disorders while adjusting for important confounders not previously considered, based on a single large database, provides a unique contribution.

## 2. Materials and Methods

### 2.1. Study Population

The intended target population is employees, aged 18–64, in the Mountain West region of the United States. Analyses are based on employees receiving health insurance from the Deseret Mutual Benefit Administrator (DMBA). The insurance provider was established in 1970 to provide health insurance and retirement income to employees and their families of the Church of Jesus Christ of Latter-day Saints. Electronic claims data were retrieved for the years 2017–2021. A “claim” is a notification to DMBA requesting a medical benefit payment. Pharmaceutical claims are not included in this study. Geographic areas represented by enrollees included Utah (74%), Idaho (9%), Pacific states (9%), and other American states (8%).

Each year the cohort of DMBA enrollees consists of approximately 27% employees, 21% spouses, 48% dependent children, and 4% other (e.g., married child, stepchild, disabled dependent). Among employees, 34% work in the Church education system (Brigham Young University [Utah, Idaho, Hawaii] and Ensign College), seminaries, and institutes; 31% as manual laborers (e.g., electricians, custodians, maintenance technicians, and farmers); 10% in other companies; 6% were retired; and the remaining 19% worked in other capacities. Retirees are not considered in this study. Employee retention was about 92% (80% in ages 18–29, 95% in ages 30–64, and 76% in ages 65 or older) from year to year during the study. The number of employees insured through DMBA dropped for individuals aged 65 and older as they became eligible for Medicare.

### 2.2. Data Collection

Data are obtained from a retrospective cohort. The study involved DMBA employees aged 18–64 in 2017–2021 (M = 21,362). These data represent eligibility data linked to automated medical claims records using a common identifying number, with no selection of subgroups to bias the results. All employees in this data were eligible for comprehensive insurance coverage. DMBA gives mental health services and related benefits to all medical plans, so there is likely no confounding variable regarding mental disorders. A mental health diagnosis under DMBA does not lead to losing insurance; under-reporting is not likely. After linking the data and prior to analysis, the database was de-identified according to Health Insurance Portability and Accountability Act (HIPAA) guidelines. The need for ethical approval and consent was waived by the institutional review board at Brigham Young University (IRB2021-157).

International Classification of Diseases, Tenth Revision, Clinical Modification (ICD-10-CM) was used for medical billing of the conditions considered in this study [44]. The codes used for classifying the conditions of interest are R47 for speech disturbances, R49 for voice and resonance disorders, R05 for cough, and R06 for abnormalities of breathing.

R47—Speech Disturbances: Dysphasia and aphasia, dysarthria and anarthria, slurred speech, fluency disorder or conditions classified elsewhere, other speech disturbances, and unspecified speech disturbances.

R49—Voice Disorders: Dysphonia, aphonia, hypernasality, hypo-nasality, other voice and resonance disorders, and unspecified voice and resonance disorder.

R05—Cough: Acute cough, subacute cough, chronic cough, cough syncope, other specified cough, cough unspecified.

R06—Breathing Abnormalities: Dyspnea, stridor, wheezing, periodic breathing, hyperventilation, mouth breathing, hiccough, sneezing, other abnormalities of breathing, unspecified abnormalities of breathing.

The Diagnostic and Statistical Manual of Mental Disorders (DSM) helps psychiatrists, physicians, clinical psychologists, and other health professionals diagnose behavioral health issues [45]. The DSM diagnostic criteria serve as a guide in determining billing codes according to the ICD-10-CM. Codes used to classify conditions relevant to the current study are F20–F29 for schizophrenia, delusional, and other non-mood-psychotic disorders (hereafter called schizophrenia); F31 for bipolar disorder; F32–F33 for depression; F40–F41 for anxiety; F42 for obsessive-compulsive disorder (OCD); F43 for stress; and F90 for ADHD. Codes used to classify sleep disorders are G470 for insomnia, G471 for hypersomnia, and G473 for sleep apnea. Other sleep disorders consist of G472 (circadian rhythm sleep disorders), G474 (narcolepsy and cataplexy), G475 (parasomnia), G476 (sleep-related movement disorders), G478 (other sleep disorders), and G479 (unspecified sleep disorders). Finally, codes used for classifying the potential confounding variables are K21 for gastroesophageal reflux disease, J45 for Asthma, J30 for vasomotor and allergic rhinitis, J32 for chronic sinusitis, and I10–I16 for hypertension.

Percentages and rates for the specific types of conditions each year consist of the number of enrollees filing one or more claims for each disorder divided by the number of enrollees. If multiple claims are filed in a year for a specific condition, it is only counted once in the numerator of the percentage or rate calculation for that year. However, an individual may contribute to more than one type of speech disturbance, voice disorder, cough, breathing abnormality, mental illness, sleep disorder, or other conditions each year.

Other variables considered in this study are age, sex, marital status, dependent children status, annual salary, and year. These variables are described in Table 1.

### 2.3. Statistical Techniques

Numbers, percentages, and means describe the variables. Rates of speech disturbances, voice disorders, cough, and breathing abnormalities are compared across the levels of sex, age, marital status, dependent children, and annual salary. Rates of mental illness and sleep disorders are presented for the four upper airway-related symptoms. Rates of speech disturbances, voice disorders, cough, and breathing abnormalities, mental illnesses, and sleep disorders are compared across the status of gastroesophageal reflux disease, asthma, allergic rhinitis, chronic sinusitis, and hypertension, adjusted for sex, age, marital status, dependent children, annual salary, and year. Rates of mental illnesses and sleep disorders are presented according to the status of the four symptoms, adjusting for the demographic variables, and then again adjusting for the demographic variables and gastroesophageal reflux disease, asthma, allergies, chronic sinusitis, and hypertension. This is performed by including the potential confounders in a Poisson multiple regression model. Statistical analyses were derived from Statistical Analysis System (SAS) software, version 9.4 (SAS Institute Inc., Cary, NC, USA, 2012).

## 3. Results

Rates of the selected speech disturbances, voice disorders, cough, and breathing abnormalities appear according to selected variables among DMBA employees in Table 1. The average number of employees each year is 21,362. Employees are most likely 40–59 years of age, married, with dependent children, and have an annual salary in the range of $50,000 to $100,000. The rate of cough is greater in women than in men. Rates of speech disturbances, voice disorders, cough, and breathing abnormalities tend to increase with age. Rates of cough and breathing abnormalities are greater in married employees. The rate of speech disturbances is greater in those with dependent children, but rates of cough and breathing abnormalities are lower in those with dependent children. The rate of cough increases with a higher annual salary whereas the rate of breathing abnormalities decreases with a higher annual salary.

The rate of medical claims for speech disturbances, voice disorders, cough, and breathing abnormalities appear in Table 2. The distribution of selected types of mental illness and sleep disorders for the four conditions is presented. For example, among employees with speech disturbances, 7.0% have stress, 21.7% have anxiety, 20.2% have depression, 10.1% have insomnia, 3.9% have hypersomnia, and 24.8% have sleep apnea. The highest percentages of stress, anxiety, depression, bipolar disorder, schizophrenia, suicidal ideation, insomnia, and other sleep disorders are associated with speech disturbances. High percentages of stress, anxiety, depression, and OCD are also seen in those with voice disorders. The highest percentages of hypersomnia and sleep apnea are associated with breathing abnormalities.

Of the selected potential confounders, hypertension is the most common, affecting 13.6% of employees (Table 3). Rates of speech disturbances, voice disorders, cough, and breathing abnormalities tend to be significantly greater for employees experiencing one or more of the potential confounders. Exceptions involve allergies and chronic sinusitis, which are not significantly associated with speech disturbances, and hypertension, which is not significantly associated with voice disorders.

Rates of mental illness and sleep disorders also tend to be significantly greater for those experiencing the potential confounders (Table 4). Gastroesophageal reflux is most consistently significantly associated with the selected mental illnesses and sleep disorders. The potential confounders are consistently significantly positively associated with anxiety, depression, ADHD, bipolar disorder, insomnia, hypersomnia, sleep apnea, and other sleep problems. There is also statistical evidence that asthma is positively associated with schizophrenia and suicidal ideation, allergies are positively associated with OCD, and chronic sinusitis is positively associated with suicidal ideation.

Rates of the types of mental illness and sleep disorders according to the status of speech disturbances, voice disorders, cough, and breathing abnormalities appear in Table 5. In the top half of the table, the rate ratios are adjusted for age, sex, marital status, dependent children, salary, and year. In the bottom half of the table, the rate ratios are adjusted for these demographic variables as gastroesophageal reflux, asthma, allergies, sinusitis, and hypertension. Significant rate ratios prior to the additional adjustments tend to remain significant. Exceptions are influenced by small numbers.

Rate ratios measuring the strength of the association between upper airway-related symptoms and any mental illness appear in Figure 1. Each rate ratio remains significantly positive, after additional adjustment for the potential confounders (i.e., the confidence intervals do not overlap 1). Adjustment for the demographic variables does not significantly change the strength of the relationship between upper airway-related symptoms and mental illness, as indicated by the overlapping confidence intervals. However, further adjustment for gastroesophageal reflux, asthma, allergies, sinusitis, and hypertension significantly lowers the rate ratio for cough, and breathing abnormalities. Asthma followed by gastroesophageal reflux had the largest effects on the rate ratios for cough and breathing abnormalities. The lower rate ratio for voice disorder is not significant because of small numbers. The adjusted rate ratios of mental illness are highest among individuals with speech disturbances and lowest for cough.

A similar graph is presented for sleep disorders (Figure 2). Each rate ratio remains significantly positive, after additional adjustment for the potential confounders (i.e., the confidence intervals do not overlap 1). Adjustment for the demographic variables lowers the strength of the relationship between upper airway-related symptoms and sleep disorders, significantly so for cough and breathing (see nonoverlapping confidence intervals). Age had the largest effect on the rate ratios. Additional adjustments for gastroesophageal reflux, asthma, allergies, sinusitis, and hypertension further lowered the rate ratios, significantly so for cough and breathing abnormalities. Hypertension had the largest effect on the rate ratio for speech disturbance; gastroesophageal reflux and asthma had the largest effects on the rate ratio for voice disorder; asthma and hypertension had the largest effects on the rate ratio for cough; and hypertension had the largest effect on the rate ratio for breathing abnormalities. Adjustments for the demographic and other variables significantly lowered the rate ratios for each of the upper airway-related symptoms. The adjusted rate ratios of sleep disorders are highest among individuals with breathing abnormalities and lowest for cough.

The severity of the upper airway-related symptoms, represented by the number of claims filed each year, is positively related to the rate of mental illness (Figure 3) and sleep disorders (Figure 4). Each of these relationships is statistically significant (Chi-square *p* < 0.0001). The increasing positive association with severity is most pronounced for speech disturbances (for mental) and cough (for sleep disorders). The increasing positive association from 1–4 claims to 10 or more claims is most noticeable for speech disturbances and cough.

## 4. Discussion

The current paper identifies and compares the level of comorbidity between upper airway-related symptoms and selected mental illnesses and sleep disorders. A strength of this study is that comparisons are made using information from a single, large database. Rate ratios measuring the strength of the association between variables were adjusted for certain demographic variables, as well as gastroesophageal reflux, asthma, allergies, sinusitis, and hypertension. A better understanding of these associations may prompt further exploration of patients’ conditions by healthcare professionals, leading to more holistic diagnoses that better reflect the needs of individual patients. Rates of speech disturbances, voice disorders, cough, and breathing abnormalities are associated with certain demographic variables. Specifically, women have higher rates of cough, which is consistent with other research [46]. Each of the conditions tended to increase with age, which is consistent with older age being related to things that increase the prevalence of these problems [47].

Married (vs. nonmarried) have higher levels of cough and abnormalities of breathing. Asthma has previously displayed this same association, with prior reasoning suggesting that the risk of a given condition is increased when a spouse has that condition [48]. Though further research is required to establish causation, speculation exists that many root-behavioral causes of certain illnesses and conditions may be more likely in an individual if they cohabitate with another who already has such behavior.

Having dependent children is associated with higher levels of speech disturbances, possibly because children lead to more intense voice use as well as common aging problems, which lead to thinner vocal cords [49]. Having dependent children is also associated with lower cough and abnormalities of breathing, possibly because those who have children are generally healthier [50].

Abnormalities of breathing tend to be lower in those with higher income, possibly because higher income is related to generally better health overall [51].

The generally positive associations between the potential confounders and the speech disturbances, voice disorders, cough, and breathing abnormalities are consistent with other studies [15,16,17,18,19,20,21,22,23,24,25,26,27,28,29,30,31]. For example, gastroesophageal reflux may contribute to voice disorders because of acid reflux as the acidity may damage voice functionality [15]. Cough and abnormalities of breathing may derive from asthma as it narrows airways [20,52]. In addition, the generally positive associations between gastroesophageal reflux, asthma, allergies, chronic sinusitis, hypertension, mental illnesses, and sleep disorders are consistent with other studies [32,33,34,35,36,37,38,39,40,41,42,43]. Hence, it is important to adjust for the potential confounding effects of gastroesophageal reflux, asthma, allergies, sinusitis, and hypertension.

The current study extends previous research by simultaneously considering several mental illnesses and sleep disorders, which allowed us to make important comparisons. In addition, although studies have shown that speech disturbances, voice disorders, cough, and breathing abnormalities are positively associated with certain mental illnesses [1,2,3,4,5,6], beyond age, sex, smoking, body mass index (BMI), and ACE Inhibitors [2,4], these studies did not adjust for the five important confounders (gastroesophageal reflux disease, asthma, allergies, chronic sinusitis, and hypertension) we considered in this study. These confounders had significant influences on lowering the rate ratios measuring the association between cough and breathing abnormalities, mental illness, and each of the upper airway-related symptoms and sleep disorders. However, all the adjusted rate ratios remained significantly positive. The adjusted rate ratios of mental illness are highest in those with speech disturbances and lowest for cough. The adjusted rate ratios of sleep disorders are highest among individuals with breathing abnormalities and lowest for cough.

The strength of the positive associations between upper airway-related symptoms and mental illness and sleep disorders increases with greater severity of the symptoms. Little previous research has shown this relationship. However, one study did find a positive association between the severity of cough and anxiety and depression [53]. The strongest positive association occurred for speech disturbances (for mental) and cough (for sleep disorders). Future research may investigate these findings.

Little attention has previously been given to the study of associations between upper airway-related symptoms and neurobiological disorders such as ADHD, bipolar disorders, OCD, and schizophrenia. The current study found that higher percentages of bipolar disorder, schizophrenia, suicidal ideation, and insomnia are associated with speech disturbances. Bipolar disorder and schizophrenia both appear to follow this trend due to manic-related symptoms [54,55]. Likely, the speech disturbances themselves derive from a thought disturbance of some kind. Suicidal ideation has not previously presented a relationship with speech disturbances; however, a previous study used acoustic properties of speech to identify cues of suicide risk [56]. This fluctuation in voice could present itself as a speech disorder.

Speech disturbances, voice disorders, cough, and breathing abnormalities are positively associated with insomnia and sleep apnea. Poor sleep quality has been previously linked to poor vocal quality [7,8], and sleep apnea previously associated with chronic cough and breathing abnormalities [9,10,11,12]. Given that insomnia leads to daytime sleepiness [57], the possibility exists that daytime sleepiness may result in various speech disturbances. Further, insufficient sleep can adversely affect the frontal lobe of the brain, which is responsible for our communication skills [7].

## 5. Limitations

The study results are based on an employee population, so generalization of the results is limited to this type of population, who are generally healthier than the nonworking population. Other factors also may limit the generalization of the results. Specifically, the study population was employees of the Church of Jesus Christ of Latter-day Saints. While not all employees are members of the Church, those who are often follow a health code of refraining from tobacco use and alcohol consumption. Further, approximately 74% of employees resided in Utah, which has unique characteristics that may be associated with the variables considered in this study (e.g., high-altitude living and low tobacco use). In addition, most individuals aged 65 years or older opted out of DMBA insurance, resulting in their exclusion from the study.

The number of employees who filed healthcare claims for speech disturbances and voice disorders was small. Assessing relationships between these cases and certain rare mental health conditions were limited by small numbers. Further, rates of upper respiratory-related symptoms, mental illness, and sleep disorders may be biased downward because less severe cases may not have sought medical care and, thus, not be represented in the results.

Not all confounders of the relationships assessed in this study are known or were considered. Only the more prominent confounders identified in other studies were included. The database did not contain information about lifestyle factors and social determinants of health, so they were not included as potential confounders.

Finally, the study design limited us to identifying statistical associations and not causal relationships.

## 6. Conclusions

Speech disturbances, voice disorders, cough, and breathing abnormalities are positively associated with mental illnesses and sleep disorders, even after adjustment for important confounding variables. The associations with mental illnesses are most consistent for stress, anxiety, and depression, with small numbers limiting the significance of the results for other mental illnesses. However, there is statistical evidence that ADHD and bipolar disorder are positively associated with cough and abnormalities of breathing. OCD is statistically associated with voice disorders and cough. Schizophrenia and suicidal ideation are most strongly linked with speech disturbances and voice disorders. Future research should explore these important associations. There are also positive associations between upper airway-related symptoms and sleep disorders. The strongest positive association involves breathing abnormalities and sleep disorders (particularly hypersomnia, sleep apnea, and other sleep disorders).

Adjustment for gastroesophageal reflux, asthma, allergies, sinusitis, and hypertension had a significant influence on lowering the rate ratios measuring the association between cough and breathing abnormalities and mental illness and each of the upper airway-related symptoms and sleep disorders. However, the fully adjusted rate ratios remained significantly positive. Future research may explore why the adjusted rate ratios of mental illness are highest in those with speech disturbances and lowest for cough and why the adjusted rate ratios of sleep disorders are highest in those with breathing abnormalities and lowest for cough. Increasing severity of upper airway-related symptoms is positively associated with mental illness and sleep disorders. Future research may explore why the association involving mental illness is greatest for those with speech disturbances and why the association involving sleep disorders is greatest for those with cough.

Understanding the findings in this study can provide a channel by which physicians can consider alternate or comorbid conditions, thereby facilitating appropriate treatment.

## Figures and Tables

**Figure 1 ijerph-20-07173-f001:**
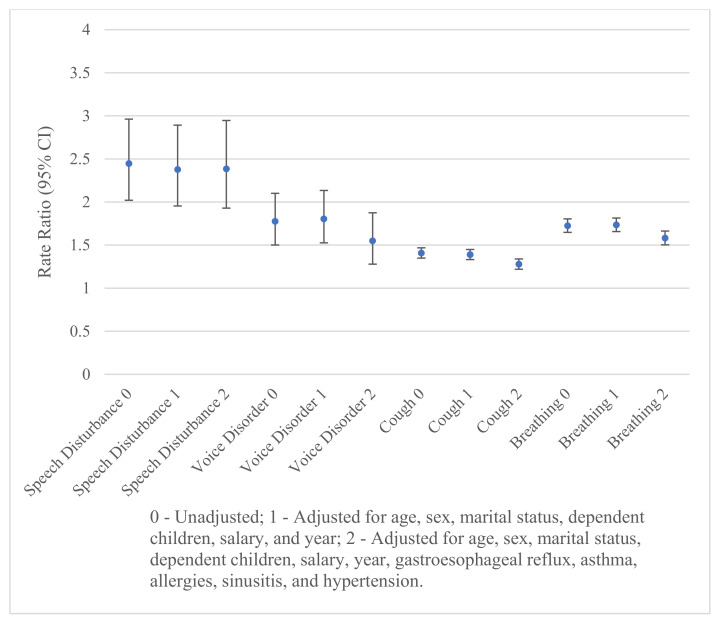
Rates of any mental illness according to speech disturbances, voice disorders, cough, and breathing abnormalities.

**Figure 2 ijerph-20-07173-f002:**
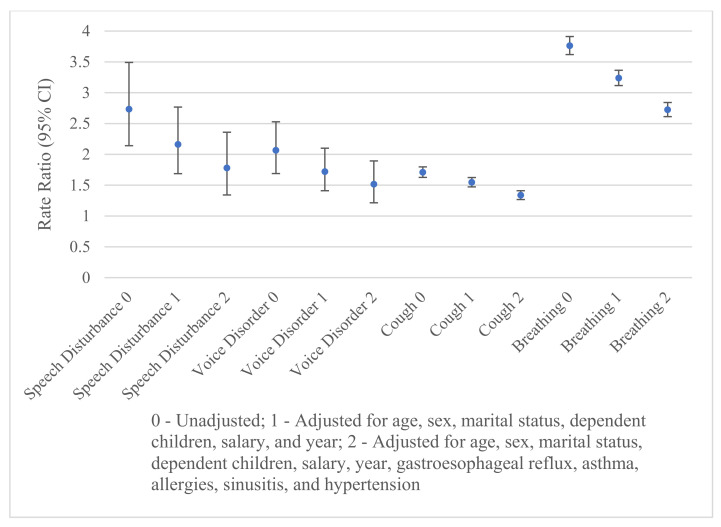
Rates of any sleep disorder according to speech disturbances, voice disorders, cough, and breathing abnormalities.

**Figure 3 ijerph-20-07173-f003:**
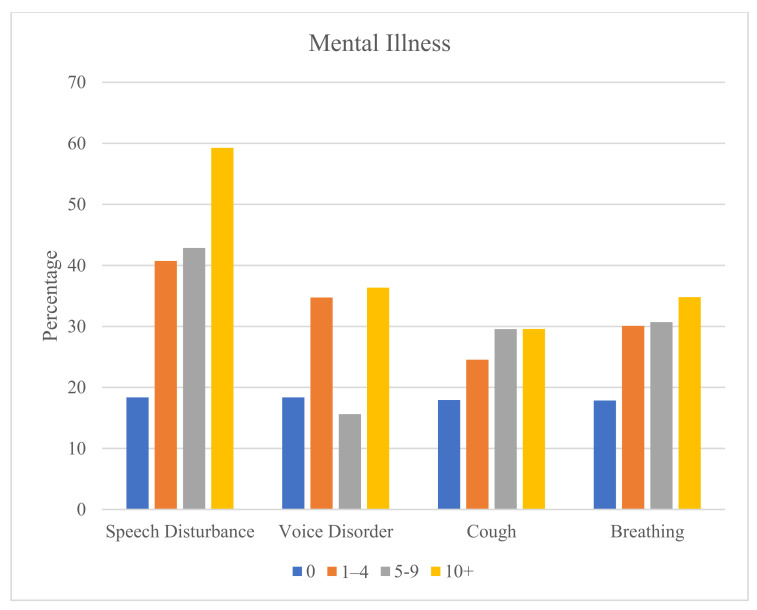
Rates (%) of any mental illness according to number of claims for upper airway-related symptoms.

**Figure 4 ijerph-20-07173-f004:**
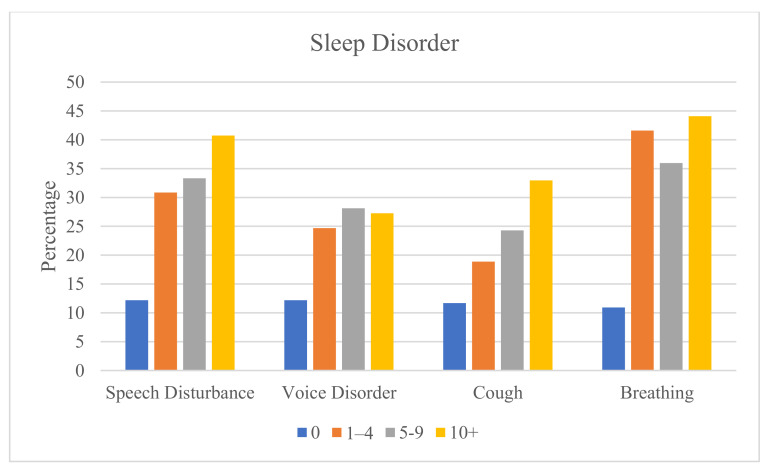
Rates (%) of any sleep disorder according to number of claims for upper airway-related symptoms.

**Table 1 ijerph-20-07173-t001:** Speech disturbances, voice disorders, cough, and abnormalities of breathing according to selected variables among DMBA employees, 2017–2021.

	No.	%	Speech Disturbances			Voice Disorders			Cough			Breathing Abnormalities		
			Rate Ratio ^†^	95% LCL ^†^	95% UCL ^†^	Rate Ratio ^†^	95% LCL ^†^	95% UCL ^†^	Rate Ratio ^†^	95% LCL ^†^	95% UCL ^†^	Rate Ratio ^†^	95% LCL ^†^	95% UCL ^†^
Sex														
Men	73,446	68.76	1.00			1.00			1.00			1.00		
Women	33,363	31.24	1.30	0.86	1.96	1.13	0.83	1.54	1.25	1.18	1.32	0.99	0.92	1.06
Age														
18–29	12,143	11.37	1.00			1.00			1.00			1.00		
30–39	22,967	21.50	1.58	0.39	6.38	1.69	0.82	3.50	1.37	1.21	1.54	1.45	1.24	1.71
40–49	27,495	25.74	4.47	1.22	16.33	3.53	1.69	7.41	1.64	1.46	1.85	2.22	1.91	2.59
50–59	28,323	26.52	5.54	1.43	21.44	4.15	1.97	8.74	2.00	1.79	2.24	2.38	2.05	2.76
60–64	15,881	14.87	15.97	4.26	59.90	4.80	2.15	10.73	2.25	1.99	2.55	2.78	2.37	3.25
Married														
No	22,808	21.35	1.00			1.00			1.00			1.00		
Yes	84,001	78.65	0.70	0.42	1.15	1.00	0.69	1.46	1.20	1.11	1.29	1.20	1.10	1.31
Dependent Children														
No	37,806	35.4	1.00			1.00			1.00			1.00		
Yes	69,003	64.6	2.01	1.25	3.23	0.90	0.66	1.22	0.94	0.88	1.00	0.90	0.83	0.97
Annual Salary														
<50 K	30,448	28.51	1.00			1.00			1.00			1.00		
50–<100 K	41,964	39.29	0.93	0.59	1.48	1.08	0.78	1.51	1.10	1.03	1.17	0.87	0.81	0.94
≥100 K	29,892	27.99	0.65	0.37	1.16	1.15	0.75	1.77	1.06	0.98	1.14	0.82	0.75	0.90
Missing	4505	4.22	1.51	0.87	2.64	1.11	0.61	2.04	0.93	0.82	1.06	1.18	1.03	1.35
Year														
2017	21,360	20.00	1.00			1.00			1.00			1.00		
2018	21,835	20.44	0.87	0.51	1.49	1.07	0.74	1.54	0.98	0.90	1.05	0.97	0.89	1.07
2019	21,663	20.28	0.97	0.58	1.62	1.14	0.79	1.64	1.00	0.93	1.08	1.10	1.00	1.20
2020	20,891	19.56	1.03	0.61	1.74	0.80	0.53	1.20	1.53	1.42	1.64	1.25	1.14	1.37
2021	21,060	19.72	0.56	0.30	1.05	1.13	0.78	1.63	1.17	1.09	1.26	1.12	1.02	1.23

^†^ Adjusted for the variables in the table.

**Table 2 ijerph-20-07173-t002:** Distribution of mental illness and sleep disorders according to speech disturbances, voice disorders, cough, and breathing abnormalities among DMBA employees, 2017–2021.

			Stress	Anx	Dep	ADHD	Bipolar	OCD	Schizo	Suicidal Ideation	Insomnia	Hypersomnia	Sleep Apnea	Other Sleep
	No.	%	%	%	%	%	%	%	%	%	%	%	%	%
All Employed	106,809	100	2.24	9.46	8.46	2.05	0.65	0.41	0.09	0.14	2.25	1.07	10.18	0.80
Speech Disturb	129	0.12	6.98	21.71	20.16	2.33	1.55	---	0.78	1.55	10.08	3.88	24.81	4.65
Voice Disorders	282	0.26	6.03	18.09	17.02	2.84	0.71	1.06	0.35	0.71	5.67	2.84	21.28	1.42
Cough	7010	6.56	3.62	13.47	12.01	2.57	1.11	0.61	0.11	0.19	3.98	2.01	16.90	1.26
Breathing	4570	4.28	3.96	17.55	15.54	2.63	1.05	0.55	0.15	0.33	5.40	6.13	36.81	4.27

ADHD: Attention deficit hyperactivity disorder. OCD: Obsessive-compulsive disorder. Note: Speech disturbances comprise dysphasia and aphasia, dysarthria and anarthria, other speech disturbances, and unspecified speech disturbances. Other sleep refers to circadian rhythms sleep disorders, narcolepsy and cataplexy, parasomnia, sleep-related movement disorders, other sleep disorders, and unspecified sleep disorders.

**Table 3 ijerph-20-07173-t003:** Speech disturbances, voice disorder, cough, and breathing abnormalities associated with selected variables for DMBA employees, 2017–2021.

			Speech Disturbances			Voice Disorders			Cough			Breathing Abnormalities		
	No.	%	Rate Ratio^†^	95% LCL ^†^	95% UCL ^†^	Odds Ratio ^†^	95% LCL ^†^	95% UCL ^†^	Rate Ratio ^†^	95% LCL ^†^	95% UCL ^†^	Rate Ratio ^†^	95% LCL ^†^	95% UCL ^†^
Gastroesophageal Reflux	4637	4.34	2.89	1.76	4.75	7.65	5.90	9.91	2.10	1.95	2.26	2.44	2.24	2.66
Asthma	3505	3.28	2.37	1.32	4.26	4.20	2.96	5.95	3.65	3.42	3.90	4.54	4.21	4.90
Allergies	4032	3.77	1.44	0.68	3.08	3.95	2.83	5.51	2.44	2.26	2.63	2.48	2.25	2.73
Chronic Sinusitis	3147	2.95	1.44	0.63	3.28	2.02	1.25	3.26	2.54	2.35	2.76	2.12	1.89	2.37
Hypertension	14,519	13.59	2.90	2.00	4.23	1.23	0.91	1.66	1.48	1.40	1.57	2.29	2.14	2.44

^†^ Adjusted for age, sex, marital status, dependent children, salary, and year. Shaded cells are statistically significant at the 0.05 level. Speech disturbances comprise dysphasia and aphasia, dysarthria and anarthria, other speech disturbances, and unspecified speech disturbances.

**Table 4 ijerph-20-07173-t004:** Mental illness and sleep disorders associated with selected variables for DMBA employees, 2017–2021.

	Stress	Anx	Dep	ADHD	Bipolar	OCD	Schizo	Suicidal Ideation	Insomnia	HypersSomnia	Sleep Apnea	Other Sleep
	Rate Ratio ^†^											
Gastroesophageal Reflux	1.93	1.76	1.72	1.44	1.90	1.58	1.82	3.29	1.77	2.07	1.73	2.29
Asthma	1.69	1.51	1.66	1.93	2.03	0.90	3.67	2.45	1.71	2.32	2.24	1.74
Allergies	1.56	1.52	1.48	1.71	1.44	2.28	2.07	1.64	1.79	1.74	1.77	1.47
Chronic Sinusitis	1.21	1.52	1.50	1.53	1.86	1.35	1.13	2.75	1.73	1.60	1.57	1.99
Hypertension	1.07	1.30	1.43	1.56	1.32	1.00	1.14	1.50	1.30	1.91	2.38	1.42

^†^ Adjusted for age, sex, marital status, dependent children, salary, and year. Shaded cells are statistically significant at the 0.05 level. Other sleep refers to circadian rhythms sleep disorders, narcolepsy and cataplexy, parasomnia, sleep-related movement disorders, other sleep disorders, and unspecified sleep disorders.

**Table 5 ijerph-20-07173-t005:** Rate of mental illness and sleep disorders according to speech disturbances, voice disorders, cough, and breathing abnormalities among DMBA employees, 2017–2021.

	Stress	Anx	Dep	ADHD	Bipolar	OCD	Schizo	Suicidal Ideation	Insomnia	Hypersomnia	Sleep Apnea	Other Sleep
	Rate Ratio ^†^											
Speech Disturb	2.70	2.30	2.20	1.33	2.58	---	1.90	7.47	3.29	3.46	1.94	5.28
Voice Disorders	2.75	1.99	1.97	1.62	1.05	3.41	4.72	6.86	2.14	2.36	1.73	1.63
Cough	1.63	1.46	1.43	1.33	1.91	1.73	1.26	1.54	1.69	1.87	1.57	1.58
Breathing	1.83	1.98	1.89	1.40	1.60	1.56	1.32	2.76	2.24	6.69	3.48	6.21
	Rate Ratio ^‡^											
Speech Disturb	2.87	2.61	2.08	1.62	1.80	---	4.44	3.23	3.18	2.62	1.46	6.27
Voice Disorders	2.95	1.60	1.81	1.54	1.13	4.11	1.48	5.46	2.05	1.42	1.48	0.81
Cough	1.59	1.36	1.30	1.20	1.78	1.69	1.00	1.31	1.50	1.62	1.34	1.54
Breathing	1.61	1.82	1.69	1.24	1.43	1.48	1.21	1.87	2.00	5.66	2.89	5.60

^†^ Adjusted for age, sex, marital status, dependent children, salary, and year. ^‡^ Adjusted for age, sex, marital status, dependent children, salary, year, gastroesophageal reflux disease, asthma, allergies, chronic sinusitis, and hypertension. Shaded cells are statistically significant at the 0.05 level. Speech disturbances comprise dysphasia and aphasia, dysarthria and anarthria, other speech disturbances, and unspecified speech disturbances. Other sleep refers to circadian rhythms sleep disorders, narcolepsy and cataplexy, parasomnia, sleep-related movement disorders, other sleep disorders, and unspecified sleep disorders.

## Data Availability

The data presented in this study are available on request from the corresponding author. The data are not publicly available due to stipulations from the DMBA.

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
