# Peer review of "Upper Airway-Related Symptoms According to Mental Illness and Sleep Disorders among Workers Employed by a Large Non-Profit Organization in the Mountain West Region of the United States"

_ijerph, 2023, doi:10.3390/ijerph20247173_

Round 1

Reviewer 1 Report

Comments and Suggestions for Authors

Dear Author, the reviewer has following comments to make about the manuscript-

TITLE

The reviewer would like to suggest modification to account for the Clarity on nature of study, Indication of specificity, Simplicity, Reducing the potential for Misinterpretation.

INTRODUCTION

1. While the introduction outlines the comorbidities and potential confounders, it does not clearly state specific hypotheses or research questions. This lack of specificity may make it difficult to understand the precise aims of the study.

2. The introduction heavily focuses on the comorbidity of physical symptoms with mental illnesses and sleep disorders. However, it does not sufficiently explore or explain the possible mechanisms or reasons behind these comorbidities.

3. While the study identifies five potential confounders, it does not clarify how these will be controlled or measured. The complexity of these confounders and their interrelationships with both the independent and dependent variables could significantly impact the study's outcomes.

4. The introduction suggests a very broad scope, examining multiple physical symptoms (speech disturbances, voice disorders, cough, and breathing abnormalities) and their association with a range of mental illnesses and sleep disorders. This broad scope might dilute the focus and make it challenging to draw specific conclusions.

5. The introduction could benefit from more context about why this research is important. For instance, understanding the clinical significance or the potential impact on treatment and management strategies for these conditions would add depth.

6.   While the introduction cites previous research, it does not clearly state the gaps or limitations in the existing literature that the current study aims to address.

7.   introduction mentions the bidirectional nature of the relationships between physical symptoms and mental health/sleep disorders but does not elaborate on how this complexity will be addressed in the study.

8.   Without information on the study population or setting, there are potential concerns about the generalizability of the findings.

9. Given the complex interplay of the conditions mentioned, there is a risk that the study might oversimplify these relationships, leading to conclusions that may not adequately reflect the intricacies involved.

MATERIAL AND MWTHOD

 1. The study population is limited to employees and their families of the Church of Jesus Christ of Latter-day Saints receiving health insurance from a specific provider. This could limit the generalizability of the findings to other populations.

2. The study does not include pharmaceutical claims, which could be a significant factor in understanding the treatment and management of the conditions being studied.

3. The majority of the study population is from Utah, with smaller percentages from other states. This geographic concentration may not represent the broader population, especially in terms of environmental or cultural factors that could influence the conditions studied.

4. The study focuses on individuals aged 18–64, excluding older adults who might have different patterns of speech disturbances, voice disorders, cough, breathing abnormalities, and mental illnesses.

5. The exclusion of retirees from the study could omit important data, especially since older age groups might have different or more pronounced symptoms.

6. The study relies on claims data, which might not capture all relevant cases, especially if individuals did not seek medical help or if their conditions were not accurately reflected in the claims.

7. While the study used standard classification systems, they have limitations in capturing the complexity and nuances of mental health and sleep disorders.

8. Although the study considers potential confounders like gastroesophageal reflux, asthma, allergies, sinusitis, and hypertension, there might be other significant confounders not accounted for.

9. The methods of statistical analysis are briefly mentioned, but there is a lack of detail on how the data will be specifically analyzed to account for the complexity and potential bidirectional nature of the associations.

10. The method of counting multiple claims for a specific condition only once per year might underestimate the prevalence or severity of the conditions.

11. While demographic variables are considered, the impact of socio-economic status, lifestyle factors, and other social determinants of health are not explicitly mentioned.

12. The method of adjusting for confounders is mentioned, but more detail on how this will be done and the potential impact on the results would be beneficial.

RESULTS

1. While the study attempts to adjust for potential confounders, there may still be unaccounted variables that could influence the results. The impact of these uncontrolled confounders on the study's conclusions is not fully addressed.

2. The results mention that exceptions in significant rate ratios are influenced by small numbers. This suggests potential issues with statistical power or sample size in certain analyses, which could affect the reliability of these findings.

3. The use of rate ratios to describe associations is informative, but it's important to note that these do not imply causality. The study does not explicitly clarify this, which might lead to misinterpretation of the results.

4. The study seems to have a complex set of interactions between the various conditions and confounders. However, the results section (text) does not thoroughly explore or explain these interactions, which could be crucial for understanding the underlying mechanisms.

 5. The study adjusts for demographic variables, but the impact of these adjustments on the associations is not deeply explored. For instance, how factors like age, sex, or salary specifically influence the associations would be valuable information.

6. The results indicate that additional adjustments for potential confounders significantly lower rate ratios for cough and breathing abnormalities. This finding warrants further exploration to understand the nature of these relationships.

7. Given that the data spans several years, it's not clear if a longitudinal analysis was conducted to understand how these associations might change over time.

8. The study groups various mental illnesses and sleep disorders together. More detailed analysis of specific conditions could provide a clearer understanding of these associations.

DISCUSIION

1. The study population is specific to employees of a particular organization with comprehensive insurance coverage. This specificity may limit the generalizability of the findings to other populations, especially those without such comprehensive insurance or from different occupational backgrounds.

2. The study acknowledges the potential for a healthy worker bias, which could skew results. Employees in regular work may generally be healthier than the general population, potentially affecting the prevalence and severity of the conditions studied.

3. The study includes only workers aged 18-64 years. This exclusion of older adults may limit the applicability of the findings to an aging population, which could have different patterns of speech disturbances, voice disorders, cough, breathing abnormalities, and mental illnesses.

4. The study did not measure the severity of the conditions, which could be a crucial factor in understanding the strength of the associations. The severity of symptoms can significantly impact the quality of life and the nature of comorbidities.

5. The reliance on medical claims data means that less serious conditions that did not result in a medical claim may be underreported. This could lead to an underestimation of the prevalence of the conditions studied.

6. The discussion acknowledges the potential inconsistency in mental health diagnoses, which could affect the reliability of the data and the conclusions drawn from it.

7. While the study discusses associations between various conditions, it does not establish causality. The discussion could benefit from a clearer distinction between correlation and causation.

8. The discussion primarily focuses on physical conditions and their associations with mental health and sleep disorders. However, it lacks a thorough exploration of non-physical factors such as psychological, social, or environmental influences.

9. Although the study mentions bidirectional relationships, there is limited discussion on how these relationships might influence the findings or the implications for treatment and management.

10. The discussion seems to focus heavily on statistical significance without adequately addressing the practical or clinical significance of the findings.

11. While the study adjusts for certain confounders, there may be other unaddressed variables that could influence the results, such as lifestyle factors, socioeconomic status, or genetic predispositions.

Author Response

Comments and Suggestions for Authors

Reviewer 1

TITLE

The reviewer would like to suggest modification to account for the Clarity on nature of study, Indication of specificity, Simplicity, Reducing the potential for Misinterpretation.

INTRODUCTION

  1. While the introduction outlines the comorbidities and potential confounders, it does not clearly state specific hypotheses or research questions. This lack of specificity may make it difficult to understand the precise aims of the study.

Response: The aim of this descriptive study was to address whether the relationship between each of the four upper airway related symptoms and selected mental illnesses and sleep disorders differs and remains statistically significant after adjusting for certain known confounders. In the second paragraph we added “Upper airway related symptoms like speech disturbances, voice disorders, cough, and breathing abnormalities are often studied together. For example, research has related each of them and their combination to quality-of-life measures [13,14]. We also see their complex comorbid relationships with mental illnesses and sleep disorders [1-12]. However, it is not known whether their relationships are similar and remain statistically significant after adjusting for certain potential confounders….”

In the final paragraph of the Introduction, we emphasize that this is a “descriptive” study.

The study is descriptive so we did not specify hypotheses, which would have required specific sample sizes to have appropriate power to test the hypotheses.

  1. The introduction heavily focuses on the comorbidity of physical symptoms with mental illnesses and sleep disorders. However, it does not sufficiently explore or explain the possible mechanisms or reasons behind these comorbidities.

Response: Reviewer 2 correctly noted that our data does not allow us to address causal mechanisms and asked that we remove mention of causality in the second paragraph of the Introduction. We have rewritten the second paragraph, as mentioned in our response to #1.

  1. While the study identifies five potential confounders, it does not clarify how these will be controlled or measured. The complexity of these confounders and their interrelationships with both the independent and dependent variables could significantly impact the study's outcomes.

Response: The final paragraph of the Introduction now says: “The purpose of the current descriptive study is to determine and compare the strength of associations between speech disturbances, voice disorders, cough, and breathing abnormalities and selected mental illnesses and sleep disorders while statistically adjusting for demographic and five important variables. The potential confounders will be shown to correlate with both the upper airways related symptoms and, independent of those relationships, correlate with the mental illness and sleep disorder variables. The study is based on employee healthcare claims data from a large non-profit organization in the Mountain West Region of the United States. While other studies have examined some of the relationships considered in this study, our comprehensive assessment of multiple upper airways related symptoms and mental illness and sleep disorders while adjusting for important confounders not previously considered, based on a single large database, provides a unique contribution.”

  1. The introduction suggests a very broad scope, examining multiple physical symptoms (speech disturbances, voice disorders, cough, and breathing abnormalities) and their association with a range of mental illnesses and sleep disorders. This broad scope might dilute the focus and make it challenging to draw specific conclusions.

Response: We try to address this concern by noting that the upper airway related symptoms are often studied together and that we need to take a broad approach so that we can assess whether the relationships between the upper airways related symptoms and mental illness and sleep disorders are similar and remain statistically significant after adjusting for certain important confounders. Doing this in a single, large database is a strength of this study.

  1. The introduction could benefit from more context about why this research is important. For instance, understanding the clinical significance or the potential impact on treatment and management strategies for these conditions would add depth.

Response: The following was added to paragraph 2 of the Introduction: “Understanding these relationships can help physicians know whether reducing mental illness and sleep disorders can come through treating the upper airway respiratory symptoms. If an underlying risk factor confounds some or all the relationship, that needs to be considered in determining appropriate treatment.”

  1. While the introduction cites previous research, it does not clearly state the gaps or limitations in the existing literature that the current study aims to address.

Response: We added the following sentence at the end of the Introduction: “While other studies have examined some of the relationships considered in this study, our comprehensive assessment of multiple upper airways related symptoms and mental illness and sleep disorders while adjusting for important confounders, based on a single large database, provides a unique contribution.”

  1. The introduction mentions the bidirectional nature of the relationships between physical symptoms and mental health/sleep disorders but does not elaborate on how this complexity will be addressed in the study.

Response: Please see response to comment #2.

  1. Without information on the study population or setting, there are potential concerns about the generalizability of the findings.

Response: The final paragraph in the Introduction now includes the sentence “… The study is based on employee healthcare claims data from a large non-profit organization in the Mountain West Region of the United States….”

The revised title is also intended to address the target population and generalizability.

An added Limitations section at the end of the Discussion addresses more fully the issue of generalizability.

  1. Given the complex interplay of the conditions mentioned, there is a risk that the study might oversimplify these relationships, leading to conclusions that may not adequately reflect the intricacies involved.

Response: We limit our conclusions to statistical relationships and not to causal relationships. Hopefully, the revised Introduction will make this clear. Also, please refer to the last sentence of the Limitations section.

MATERIAL AND METHODS

  1. The study population is limited to employees and their families of the Church of Jesus Christ of Latter-day Saints receiving health insurance from a specific provider. This could limit the generalizability of the findings to other populations.

Response: This and other limitations are now addressed in the following Limitations section that was added to the end of the Discussion.

“The study results are based on an employee population, so generalization of the results is limited to this type of population, who are generally healthier than the nonworking population. Other factors also may limit generalization of the results. Specifically, the study population were employees of the Church of Jesus Christ of Latter-day Saints. While not all employees are members of the Church, those who are often follow a health code of refraining from tobacco use and alcohol consumption. Further, approximately 74% of employees resided in Utah, which has unique characteristics that may be associated with the variables considered in this study (e.g., high altitude living and low tobacco use). In addition, most individuals aged 65 years or older opted out of DMBA insurance, resulting in their exclusion from the study.

The number of employees who filed healthcare claims for speech disturbances and voice disorders was small. Assessing relationships between these cases and certain rare mental health conditions were limited by small numbers. Further, rates of upper respiratory related symptoms and mental illness and sleep disorders may be biased downward because less severe cases may not have sought medical care and, thus, not be represented in the results.

Not all confounders of the relationships assessed in this study are known or were considered. Only the more prominent confounders identified in other studies were included. The database did not contain information about lifestyle factors and social determinants of health, so they were not included as potential confounders.

Finally, the study design limited us to identifying statistical associations and not causal relationships.”

  1. The study does not include pharmaceutical claims, which could be a significant factor in understanding the treatment and management of the conditions being studied.

Response: See the added Limitations section.

  1. The majority of the study population is from Utah, with smaller percentages from other states. This geographic concentration may not represent the broader population, especially in terms of environmental or cultural factors that could influence the conditions studied.

Response: See our response to #8, above.

  1. The study focuses on individuals aged 18–64, excluding older adults who might have different patterns of speech disturbances, voice disorders, cough, breathing abnormalities, and mental illnesses.

Response: See the added Limitations section.

  1. The exclusion of retirees from the study could omit important data, especially since older age groups might have different or more pronounced symptoms.

Response: See the added Limitations section.

  1. The study relies on claims data, which might not capture all relevant cases, especially if individuals did not seek medical help or if their conditions were not accurately reflected in the claims.

Response: See the added Limitations section.

  1. While the study used standard classification systems, they have limitations in capturing the complexity and nuances of mental health and sleep disorders.

Response: See the added Limitations section.

  1. Although the study considers potential confounders like gastroesophageal reflux, asthma, allergies, sinusitis, and hypertension, there might be other significant confounders not accounted for.

Response: See the added Limitations section.

  1. The methods of statistical analysis are briefly mentioned, but there is a lack of detail on how the data will be specifically analyzed to account for the complexity and potential bidirectional nature of the associations.

Response: See response to #2 under Introduction and the added Limitations section.

  1. The method of counting multiple claims for a specific condition only once per year might underestimate the prevalence or severity of the conditions.

Response: This is a standard approach taken using databases where multiple claims may result in a year. This way we can calculate the annual incidence rate. However, as noted the frequency of the claims may be a marker for severity. Two paragraphs and two figures are added to the Results, which address how severity positively relates to mental illness and sleep disorders.

  1. While demographic variables are considered, the impact of socio-economic status, lifestyle factors, and other social determinants of health are not explicitly mentioned.

Response: See the added Limitations section.

  1. The method of adjusting for confounders is mentioned, but more detail on how this will be done and the potential impact on the results would be beneficial.

Response: The following sentence was added to the Statistical Techniques section: “This is done by including the potential confounders in a Poisson multiple regression model.”

RESULTS

  1. While the study attempts to adjust for potential confounders, there may still be unaccounted variables that could influence the results. The impact of these uncontrolled confounders on the study's conclusions is not fully addressed.

Response: The following paragraph is now in the Limitations section: “Not all confounders of the relationships assessed in this study are known or were considered. Only the more prominent confounders identified in other studies were included. The database did not contain information about lifestyle factors and social determinants of health, so they were not included as potential confounders.”

  1. The results mention that exceptions in significant rate ratios are influenced by small numbers. This suggests potential issues with statistical power or sample size in certain analyses, which could affect the reliability of these findings.

Response: Please see our response to your first comment under Introduction.

  1. The use of rate ratios to describe associations is informative, but it's important to note that these do not imply causality. The study does not explicitly clarify this, which might lead to misinterpretation of the results.

Response: Please see our response to #2 and #3 under Introduction.

  1. The study seems to have a complex set of interactions between the various conditions and confounders. However, the results section (text) does not thoroughly explore or explain these interactions, which could be crucial for understanding the underlying mechanisms.

Response: Tables 3 and 4 were meant to show that because gastroesophageal reflux, asthma, allergies, chronic sinusitis, and hypertension are related to the upper airway related symptoms and, also, related to the mental illness and sleep disorders, they are confounders of the relationships between the upper airway related symptoms and mental illness and sleep disorders. We also consider whether these confounding factors also modify the relationships between the upper airway related symptoms and mental illness and sleep disorders (Table 5).

Please see our two new paragraphs describing Figures 1 and 2.

  1. The study adjusts for demographic variables, but the impact of these adjustments on the associations is not deeply explored. For instance, how factors like age, sex, or salary specifically influence the associations would be valuable information.

Response: Figure 1 has been modified to show the effect of age, sex, marital status, and salary on the rate ratios measuring the strength of the association between the upper airway related symptoms and mental illness. A second figure has been added to show this for sleep disorders. The text has been modified accordingly.

  1. The results indicate that additional adjustments for potential confounders significantly lower rate ratios for cough and breathing abnormalities. This finding warrants further exploration to understand the nature of these relationships.

Response: Please see the paragraph describing Figures 1 and 2.

Given that the data spans several years, it's not clear if a longitudinal analysis was conducted to understand how these associations might change over time.

Response: No, we did not consider the longitudinal nature of the data in this paper.

  1. The study groups various mental illnesses and sleep disorders together. More detailed analysis of specific conditions could provide a clearer understanding of these associations.

Response: This was the purpose of Table 5. Figure 1 was intended to evaluate the effect of confounding. This is explored further (see responses to #5 and #6 under Results). Because of small numbers for certain combinations, we combined the mental illnesses and sleep disorders in the figure.

DISCUSION

  1. The study population is specific to employees of a particular organization with comprehensive insurance coverage. This specificity may limit the generalizability of the findings to other populations, especially those without such comprehensive insurance or from different occupational backgrounds.

Response: This limitation is described in the newly added Limitations section.

  1. The study acknowledges the potential for a healthy worker bias, which could skew results. Employees in regular work may generally be healthier than the general population, potentially affecting the prevalence and severity of the conditions studied.

Response: We moved the limitations listed at the end of the Discussion to a more comprehensive list of limitations in the newly added Limitations section.

  1. The study includes only workers aged 18-64 years. This exclusion of older adults may limit the applicability of the findings to an aging population, which could have different patterns of speech disturbances, voice disorders, cough, breathing abnormalities, and mental illnesses.

Response: This limitation is mentioned in the Limitations section.

  1. The study did not measure the severity of the conditions, which could be a crucial factor in understanding the strength of the associations. The severity of symptoms can significantly impact the quality of life and the nature of comorbidities.

Response: The following was added at the end of the Results: “The severity of the upper airway related symptoms, represented by number of claims filed each year, is positively related to the rate of mental illness (Figure 3) and sleep disorders (Figure 4). Each of these relationships is statistically significant (Chi-square p < 0.0001). The increasing positive association with severity is most pronounced for speech disturbances (for mental) and cough (for sleep disorders). The increasing positive association from 1-4 claims to 10 or more claims is most noticeable for speech disturbances and cough.”

Please refer to the two new figures.

  1. The reliance on medical claims data means that less serious conditions that did not result in a medical claim may be underreported. This could lead to an underestimation of the prevalence of the conditions studied.

Response: Please see the added Limitations section.

  1. The discussion acknowledges the potential inconsistency in mental health diagnoses, which could affect the reliability of the data and the conclusions drawn from it.

Response: Correct.

  1. While the study discusses associations between various conditions, it does not establish causality. The discussion could benefit from a clearer distinction between correlation and causation.

Response: We tried to make this distinction clear in the Introduction and mention it again in the Limitations section.

  1. The discussion primarily focuses on physical conditions and their associations with mental health and sleep disorders. However, it lacks a thorough exploration of non-physical factors such as psychological, social, or environmental influences.

Response: Correct. This information was not available in our claims data.

  1. Although the study mentions bidirectional relationships, there is limited discussion on how these relationships might influence the findings or the implications for treatment and management.

Response: Please see our first couple responses under Introduction. 

  1. The discussion seems to focus heavily on statistical significance without adequately addressing the practical or clinical significance of the findings.

Response: Please see response #5 under Introduction. We also add at the end of the first paragraph in the Introduction “A better understanding of these associations may prompt further exploration of patients’ conditions by healthcare professionals, leading to more holistic diagnoses that better reflect the needs of individual patients.”

  1. While the study adjusts for certain confounders, there may be other unaddressed variables that could influence the results, such as lifestyle factors, socioeconomic status, or genetic predispositions.

Response: Please see the added Limitations section.

Reviewer 2 Report

Comments and Suggestions for Authors

Dear authors, is a work that involves a lot of data and variables, which offers a tempting view for the audience of the journal, however I suggest the following aspects to facilitate the understanding of the manuscript: 

In summary, it is important to point out what design, nor is the justification about the confusors in the study clear. They are not mentioned in the summary. I suggest following a summary which indicates what is not known to raise the objective and the way in which it was answered. Indicating the results and association values obtained. 

Introduction: The selected design does not allow to evaluate causality relationships, revise line 37 to 58 affirmation and adjust according to the scope of the study. It is also suggested to extend on statistics of prevalence of health problems studied in the world and in the states of the United States where sampling is taken because the prevalence is mentioned in the results. 

methodology 

Specify sample size justification   (avoid type 1 or type 2 errors)

Specify target population  (The target or reference population is the overall population that the research is directed towards.) 

Ethical aspects are not mentioned, they can be indicated in the methodology (I know that it appears at the end). 

Need The sample selection process is important in determining to what extent the results of the study are generalizable to the target population

Specify whether Selection bias can occur if every unit in the sample frame doesn’t have an equal chance of been included in the final study

specify Measurement Validity & Reliability 

The reproducibility of data processing, can be refined, was difficult for me to understand how data is obtained and analyzed. 

I didn't find the limitations of the study. There is an expectation that the researcher discusses selection biases and takes these into account when interpreting the results of the study. This also gives a clear view of whether the researcher has an overall understanding of the study design. 

Given that confusors are the central point of the work, it is important to clarify the selection of them, for analysis, as only one publication is mentioned to support their relationship with the variable of interest. Therefore, expand with more studies in the introduction line 45 to 53, on the support that these variables are confusing. 

The confidence intervals for breathing 1 and 2 are the only ones that are not intercepted in sleep problems, however, this is not discussed or mentioned in the results (figura 1). 

The text in the results describes the information of the tables, I suggest explicitly the message about the associations obtained, to indicate what was found and what is the coefficient that supports that idea. Therefore, explain the results obtained in response to the purpose of the study. 

The study of the role of confusors in the strength of the association is not discussed. The discussion should deal with the purpose of the study "The purpose of the current study is to determine and compare the strength of associ- 55 ations between speech disturbances, voice disorders, cough, and breathing abnormalities 56 and selected mental illnesses and sleep disorder while understanding the role of poten- 57 tial confounders."

Study limitations are not discussed. 

Author Response

Reviewer 2

In summary, it is important to point out what design, nor is the justification about the confusors in the study clear. They are not mentioned in the summary. I suggest following a summary which indicates what is not known to raise the objective and the way in which it was answered. Indicating the results and association values obtained. 

 Response: Is this in reference to the Abstract. We have rewritten it in response to comments made here, as follows:

“Abstract: The relationships between selected upper airway related symptoms (speech disturbances, voice disorders, cough, and breathing abnormalities) and mental illness and sleep disorders have been previously demonstrated. However, these relationships have not been compared in a single study with consideration of potential confounding variables. The current research incorporates a descriptive study design of medical claims data for employees (~21,362 per year 2017-2021) with corporate insurance to evaluate the strength of these relationships, adjusting for demographic variables and other important confounders. The upper airway related symptoms are each significantly and positively associated with several mental illness and sleep disorders after adjusting for demographic and other potential confounders. The rate of any mental illness is 138% (95% CI 93%-195%) higher for speech disturbances, 55% (95% CI 28%-88%) higher for voice disorders, 28% (95% CI 22%-34%) higher for cough, and 58% (95% CI 50%-66%) higher for breathing abnormalities, after adjustment for the confounding variables. Confounding had significant effects on the rate ratios involving cough and breathing abnormalities. The rate of any sleep disorder is 78% (95% CI 34%-136%) higher for speech disturbances, 52% (95% CI 21%-89%) higher for voice disorders, 34% (95% CI 27%-41%) higher for cough, and 172% (95% CI 161%-184%) higher for breathing abnormalities, after adjustment for the confounding variables. Confounding had significant effects on each of the upper airway related symptoms. Rates of mental illness and sleep disorders are positively associated with number of claims filed for each of the upper airway related symptoms. The comorbid nature of these conditions should guide clinicians in providing more effective treatment plans that ultimately yield the best outcome for patients.”

Introduction: The selected design does not allow to evaluate causality relationships, revise line 37 to 58 affirmation and adjust according to the scope of the study. It is also suggested to extend on statistics of prevalence of health problems studied in the world and in the states of the United States where sampling is taken because the prevalence is mentioned in the results. 

 Response: We deleted the sentence “The causal relationships between these conditions and mental illness and sleep disorders are complex, often indirect, and bidirectional.”

The paragraph has been refocused to justify why we are considering each of the four upper airway related symptoms and that we will address relationships and not causality.

The mention of “prevalence” in the Results is in the title of Table 2, where we show the distribution of mental illness and sleep disorders according to the upper airways related symptoms. The word “prevalence” should be “distribution”. We have replaced it because our study does not involve cross-sectional survey data, but rather incidence data.

Paragraph 2 now says: “Upper airway related symptoms like speech disturbances, voice disorders, cough, and breathing abnormalities are often studied together. For example, research has related each of them and their combination to quality-of-life measures [13,14]. We also see their complex comorbid relationships with mental illnesses and sleep disorders [1-12]. However, it is not known whether their relationships are similar and remain statistically significant after adjusting for certain potential confounders. Understanding these relationships can help physicians know whether reducing mental illness and sleep disorders can come through treating the upper airway respiratory symptoms. If an underlying risk factor confuses some or all the relationship, that needs to be considered in determining appropriate treatment.”

methodology 

 Specify sample size justification (avoid type 1 or type 2 errors)

Response: The sample size is based on those employees in the DMBA database who file healthcare claims. We wanted to see if the relationship between each of the four upper airway related symptoms and selected mental illnesses and sleep disorders remained statistically significant and differed after adjusting for potential confounders. Because this is a descriptive study there are no specific hypotheses that require given sample sizes to be appropriately powered.

Specify target population (The target or reference population is the overall population that the research is directed towards.) 

Response: The first sentence of the Materials and Methods section now says “The intended target population are employees, aged 18-64, in the Mountain West region of the United States.”

Ethical aspects are not mentioned, they can be indicated in the methodology (I know that it appears at the end). 

Response: We added the following to the Materials and Methods section: “After linking the data and prior to analysis, the database was de-identified according to Health Insurance Portability and Accountability Act (HIPAA) guidelines. The need for ethical approval and consent were waived by the institutional review board at Brigham Young University (IRB2021-157).”

Need The sample selection process is important in determining to what extent the results of the study are generalizable to the target population.

Response: A Limitations section was added, which says:

Limitations

“The study results are based on an employee population, so generalization of the results is limited to this type of population, who are generally healthier than the nonworking population. Other factors also may limit generalization of the results. Specifically, the study population were employees of the Church of Jesus Christ of Latter-day Saints. While not all employees are members of the Church, those who are often follow a health code of refraining from tobacco use and alcohol consumption. Further, approximately 74% of employees resided in Utah, which has unique characteristics that may be associated with the variables considered in this study (e.g., high altitude living and low tobacco use). In addition, most individuals aged 65 years or older opted out of DMBA insurance, resulting in their exclusion from the study.

The number of employees who filed healthcare claims for speech disturbances and voice disorders was small. Assessing relationships between these cases and certain rare mental health conditions were limited by small numbers. Further, rates of upper respiratory related symptoms and mental illness and sleep disorders may be biased downward because less severe cases may not have sought medical care and, thus, not be represented in the results.

Not all confounders of the relationships assessed in this study are known or were considered. Only the more prominent confounders identified in other studies were included. The database did not contain information about lifestyle factors and social determinants of health, so they were not included as potential confounders.

Finally, the study design limited us to identifying statistical associations and not causal relationships.”

Specify whether Selection bias can occur if every unit in the sample frame doesn’t have an equal chance of been included in the final study.

Response: Please refer to the added Limitations section.

specify Measurement Validity & Reliability 

Response: Please refer to the added Limitations section.

The reproducibility of data processing, can be refined, was difficult for me to understand how data is obtained and analyzed. 

Response: The first sentence under section 2.2 Data Collection says “The study involved DMBA employees aged 18–64 in 2017-2021 (M=21,362). These data represent eligibility data linked to automated medical claims records using a common identifying number.”

Also, in section 2.3 Statistical techniques we added that calculation of rate ratios, adjusted for potential confounders, using Poisson multiple regression models.

I didn't find the limitations of the study. There is an expectation that the researcher discusses selection biases and takes these into account when interpreting the results of the study. This also gives a clear view of whether the researcher has an overall understanding of the study design. 

 Response: The final couple of paragraphs in the Discussion mentioned limitations but we have put them and added to what was previously said in the new Limitations section at the end of the Discussion.

Given that confusors are the central point of the work, it is important to clarify the selection of them, for analysis, as only one publication is mentioned to support their relationship with the variable of interest. Therefore, expand with more studies in the introduction line 45 to 53, on the support that these variables are confusing. 

Response: The paragraph has been modified to now say: “A potential confounder is a common risk factor for both the exposure and outcome variables. Controlling for potential confounders is important for establishing the existence and strength of association between upper airway related symptoms and mental illness and sleep disorders. In the current study we consider five potential confounders (i.e., gastroesophageal reflux, asthma, allergies, sinusitis, and hypertension) of the associations between upper airway related symptoms and mental illness and sleep disorders. These potential confounders are considered because of their association with both the upper airway related symptoms and mental illness and sleep disorders. The studies showing relationships between these variables [1-12] did not adjust for these potential confounders, except in the study showing an association between chronic cough and mental health disturbances [4], where they adjusted for ACE inhibitors (treatment for hypertension). Previous research has shown that speech disturbances, voice disorders, cough, and breathing abnormalities are associated with gastroesophageal reflux [13-15], asthma [16-19], allergies [20-23], sinusitis [24-26], and hypertension [27-29]. In addition, gastroesophageal reflux is associated with anxiety and depression [30]; asthma is associated with ADHD, anxiety, and major depression [31-32]; allergies are associated with psychiatric disorders [33-34]; sinusitis is associated with depression [35]; hypertension is associated with anxiety, depression, impulsive eating disorders and substance use disorders [36]; and gastroesophageal reflux, asthma, allergies, sinusitis, and hypertension are each associated with sleep disorders [37-41].”

The confidence intervals for breathing 1 and 2 are the only ones that are not intercepted in sleep problems, however, this is not discussed or mentioned in the results (figura 1). 

 Response: Correct. Where the confidence intervals do not overlap, there is a significant difference. If they overlap the rate ratio in the comparison group, they are not significantly different. If they overlap but do not contain the rate ratio in the comparison group, we are unsure if there is a significant difference, and an additional test is required to determine this.

Based on comments from other reviewers, two figures were created from this figure and the additional unadjusted rate ratios (and corresponding confidence intervals) were added (see new Figures 1 and 2). The paragraphs describing these chapters are as follows, with incorporation of the confidence intervals (per your suggestion).

“Rate ratios measuring the strength of the association between upper airway related symptoms and any mental illness appear in Figure 1. Each rate ratio remains significantly positive, after additional adjustment for the potential confounders (i.e., the confidence intervals do not overlap 1). Adjustment for the demographic variables does not significantly change the strength of the relationship between the upper airway related symptoms and mental illness, as indicated by the overlapping confidence intervals. However, further adjustment for gastroesophageal reflux, asthma, allergies, sinusitis, and hypertension significantly lowers the rate ratio for cough, and breathing abnormalities. Asthma followed by gastroesophageal reflux had the largest effects on the rate ratios for cough and breathing abnormalities (data not shown). The lower rate ratio for voice disorder is not significant because of small numbers. The adjusted rate ratios of mental illness are highest among individuals with speech disturbances and the rates of sleep disorders are highest among individuals with breathing abnormalities.”

“A similar graph is presented for sleep disorders (Figure 2). Each rate ratio remains significantly positive, after additional adjustment for the potential confounders (i.e., the confidence intervals do not overlap 1). Adjustment for the demographic variables lowers the strength of the relationship between the upper airway related symptoms and sleep disorders, significantly so for cough and breathing (see nonoverlapping confidence intervals). Age had the largest effect on the rate ratios (data not shown). Additional adjustment for gastroesophageal reflux, asthma, allergies, sinusitis, and hypertension further lowered the rate ratios, significantly so for cough and breathing abnormalities. Hypertension had the largest effect on the rate ratio for speech disturbance; gastroesophageal reflux and asthma had the largest effects on the rate ratio for voice disorder; asthma and hypertension had the largest effects on the rate ratio for cough; and hypertension had the largest effect on the rate ratio for breathing abnormalities (data not shown). Adjustment for the demographic and other variables significantly lowered the rate ratios for each of the upper airway related symptoms. The adjusted rate ratios of sleep disorders are highest among individuals with breathing abnormalities and lowest for cough.”

The text in the results describes the information of the tables, I suggest explicitly the message about the associations obtained, to indicate what was found and what is the coefficient that supports that idea. Therefore, explain the results obtained in response to the purpose of the study. 

Response: We believe this has now been addressed in the two paragraphs listed in the previous response. If not, please specify what more you think is needed.

The study of the role of confusors in the strength of the association is not discussed. The discussion should deal with the purpose of the study "The purpose of the current study is to determine and compare the strength of associations between speech disturbances, voice disorders, cough, and breathing abnormalities and selected mental illnesses and sleep disorder while understanding the role of potential confounders."

Response: The following paragraph was added to the Discussion: “The current study extends previous research by simultaneously considering several mental illnesses and sleep disorders, which allowed us to make important comparisons. In addition, although studies have shown that speech disturbances, voice disorders, cough, and breathing abnormalities are positively associated with certain mental illnesses [1-6], beyond age, sex, smoking, BMI, and ACE Inhibitors [2,4], these studies did not adjust for the five important confounders (gastroesophageal reflux disease, asthma, allergies, chronic sinusitis, and hypertension) we considered in this study. These confounders had significant influences on lowering the rate ratios measuring the association between cough and breathing abnormalities and mental illness and each of the upper airway related symptoms and sleep disorders. However, all the adjusted rate ratios remained significantly positive.  The adjusted rate ratios of mental illness are highest in those with speech disturbances and lowest for cough. The adjusted rate ratios of sleep disorders are highest among individuals with breathing abnormalities and lowest for cough.”

The Conclusion also better ties the results with the intended purpose of the study.

Study limitations are not discussed. 

Response: We have added a Limitation section, as already mentioned.

Reviewer 3 Report

Comments and Suggestions for Authors

This is a review of a manuscript titled “Speech Disturbances, Voice Disorders, Cough, and Breathing Abnormalities Are Comorbid with Mental Illnesses and Sleep Disorders: What Role Does Confounding Play?”. Your article is well-written and presents important information about relationships between disorders that haven’t been extensively researched to help clarify confounding information about voice disorders, sleep disorders and mental illnesses and show a new panorama. I have some recommendations for improving specific details.

Best regards,

Title. Considering that you are working mainly with a sample corresponding to specific geographic areas (Utah enrollees 74%), I suggest including this in the title to give precise information to the readers.

I also recommend shortening the title, as in your article: “Sleep disorders related to index and comorbid mental disorders and psychotropic drugs”.

Abstract. Please include relevant statistical information.

Line 51. Considering that you have been including the references from which you study the disorders, it would be helpful that you indicate which ones are referring to eating disorders and substance abuse.

Line 71. Please include a brief job description about the education system and manual labor.

Lines 140 to 146. The rates can be presented visually in a better format if you include them in a table.

Line 152. The distribution of employees with speech disturbances could be an effect of their type of job. Could you include a table related to the job?

Figure 1. Legend 1 and Legend 2 must be under the variables.

Discussion: Does the geographical characteristic of the place of living of your sample affect the developing of speech disturbances, voice disorders, cough, and breathing abnormalities?

Author Response

Reviewer 3

Comments and Suggestions for Authors

This is a review of a manuscript titled “Speech Disturbances, Voice Disorders, Cough, and Breathing Abnormalities Are Comorbid with Mental Illnesses and Sleep Disorders: What Role Does Confounding Play?”. Your article is well-written and presents important information about relationships between disorders that haven’t been extensively researched to help clarify confounding information about voice disorders, sleep disorders and mental illnesses and show a new panorama. I have some recommendations for improving specific details.

Best regards,

Title. Considering that you are working mainly with a sample corresponding to specific geographic areas (Utah enrollees 74%), I suggest including this in the title to give precise information to the readers.

I also recommend shortening the title, as in your article: “Sleep disorders related to index and comorbid mental disorders and psychotropic drugs”.

Response: The title has been revised to say “Upper Airway Related Symptoms Related to Mental Illness and Sleep Disorders among Workers Employed by a Large Non-Profit Organization in the Mountain West Region of the United States”

Abstract. Please include relevant statistical information.

Response: The Abstract has been rewritten. Results included are: “…The rate of any mental illness is 138% (95% CI 93%-195%) higher for speech disturbances, 55% (95% CI 28%-88%) higher for voice disorders, 28% (95% CI 22%-34%) higher for cough, and 58% (95% CI 50%-66%) higher for breathing abnormalities, after adjustment for selected variables. Confounding had significant effects on the rate ratios involving cough and breathing abnormalities. The rate of any sleep disorder is 78% (95% CI 34%-136%) higher for speech disturbances, 52% (95% CI 21%-89%) higher for voice disorders, 34% (95% CI 27%-41%) higher for cough, and 172% (95% CI 161%-184%) higher for breathing abnormalities, after adjustment …”

Line 51. Considering that you have been including the references from which you study the disorders, it would be helpful that you indicate which ones are referring to eating disorders and substance abuse.

Response: Reference 36 (now 38 with the addition of two references and renumbering in the revised version) includes eating disorders and substance use disorders. For example, the Conclusion in the Abstract for that paper says “Depression, anxiety, impulsive eating disorders, and substance use disorders were significantly associated with the subsequent diagnosis of hypertension.”

Line 71. Please include a brief job description about the education system and manual labor.

Response: We now say “… Church education system (Brigham Young University [Utah, Idaho, Hawaii] and Ensign College), seminaries, and institutes; 31% as manual laborers (e.g., electricians, custodians, maintenance technicians, and farmers); …”

Lines 140 to 146. The rates can be presented visually in a better format if you include them in a table.

Response: Rates (%) appear in Table 1 (Column 3). Rate ratios in the table show, for example, that the rate of cough for women is 25% greater than for men. This result is statistically significant because the confidence interval does not overlap 1.

Line 152. The distribution of employees with speech disturbances could be an effect of their type of job. Could you include a table related to the job?

Response: It would be very interesting to compare speech disturbances by the job types, but the number of speech disturbances is 129. Breaking this down by job type to look at comorbid mental illnesses and sleep disorders is limited by sample size. Also, our broad classification of workers does not allow us to identify their specific use of their voice. For example, positions in the universities and college range from heaving voice use while teaching in the classroom to less voice use in research or administrative responsibilities. 

Should we include this in the Limitations section?

Figure 1. Legend 1 and Legend 2 must be under the variables.

Response: The legends have been moved below the variables.

Discussion: Does the geographical characteristic of the place of living of your sample affect the developing of speech disturbances, voice disorders, cough, and breathing abnormalities?

Response: The challenge is that there are a variety of job types in some of the geographic areas. Hence, differences in speech disturbances, voice disorders, cough, and breathing abnormalities among the geographic areas may be due to the environment, job type, or other factors which our data does not allow us to determine. Hence, while there may or may not be differences detected, it is impossible to know why the differences exist. An assumption in this study is that the associations measured between these variables and mental illnesses and sleep disorders will be similar across the job types and geographic areas.

Reviewer 4 Report

Comments and Suggestions for Authors

Dear authors, I really appreciate your work and efforts but there are some steps that need to be clarified, in order to proceed to approval:
a) I did not find in the text, clearly written, the opinion of the ethics committee, and a detailed description of the research protocol;
b) you refer in the text to ICD and DSM in the versions prior to the current ones, and therefore you need to adapt and modify the text in accordance with the current versions (ICD-11 and DSM-5-TR);
c) you need to better clarify, both in the introduction and in the discussions, the role of physical comorbidities with respect to psychiatric symptomatology, and borderline forms (e.g., nervous cough), and not just list them with footnotes mentioning those issues;
d) I would implement a clear distinction between neurobiological disorders and psychiatric disorders (the text lacks this element and the reading suffers greatly from this);
e) a table of data, in the discussion, would help the understanding of the text;
f) you need to detach from the discussions and conclusions, and insert them in detail, the paragraph of study limitations (in detail), and future prospects.

Author Response

Reviewer 4

Dear authors, I really appreciate your work and efforts but there are some steps that need to be clarified, in order to proceed to approval:
a) I did not find in the text, clearly written, the opinion of the ethics committee, and a detailed description of the research protocol;

Response: The Institutional Review Board Statement appears after the Conclusions, prior to the References. However, we now have also added an ethics statement to the Methods: “After linking the data and prior to analysis, the database was de-identified according to Health Insurance Portability and Accountability Act (HIPAA) guidelines. The need for ethical approval and consent were waived by the institutional review board at Brigham Young University (IRB2021-157).”

In addition, we now clarify that this is a descriptive study based on healthcare claims data. Please refer to the revised paragraphs 2 and 4 of the Introduction.

  1. b) you refer in the text to ICD and DSM in the versions prior to the current ones, and therefore you need to adapt and modify the text in accordance with the current versions (ICD-11 and DSM-5-TR);

Response: The data provided to us from DMBA was coded using ICD-10. However, reference 45 (was 43 prior to our adding two references in the revision) now refers to the current version of DSM-5-TR: “American Psychiatric Association. Diagnostic and statistical manual of mental disorders (5th ed., text rev.), 2022. https://doi.org/10.1176/appi.books.9780890425787”

  1. c) you need to better clarify, both in the introduction and in the discussions, the role of physical comorbidities with respect to psychiatric symptomatology, and borderline forms (e.g., nervous cough), and not just list them with footnotes mentioning those issues;

Response: We are limited by the ICD classifications. However, we have tried to better capture the severity of the upper airway related symptoms and mental illness and sleep disorders (please refer to the final paragraph of the Results and Figures 3 and 4). We do mention in the newly added Limitations that less severe conditions may not be reflected in our study because people with these conditions do not seek healthcare assistance.

  1. d) I would implement a clear distinction between neurobiological disorders and psychiatric disorders (the text lacks this element and the reading suffers greatly from this);

Response: Lines 323-326 now say “Little attention has previously been given to the study of associations between upper airway related symptoms and neurobiological disorders such as ADHD, bipolar disorders, OCD, schizophrenia, and suicidal ideation. The current study …”

If the reviewer could give us more guidance on what they want here, that would be greatly appreciated.

  1. e) a table of data, in the discussion, would help the understanding of the text;

Response: we are unclear how to respond to this comment. It is rare to see table summaries in the Discussion. Perhaps the reviewer can clarify this comment.

  1. f) you need to detach from the discussions and conclusions, and insert them in detail, the paragraph of study limitations (in detail).

Response: A final Limitations section was added, which says: The study results are based on an employee population, so generalization of the results is limited to this type of population, who are generally healthier than the nonworking population. Other factors also may limit generalization of the results. Specifically, the study population were employees of the Church of Jesus Christ of Latter-day Saints. While not all employees are members of the Church, those who are often follow a health code of refraining from tobacco use and alcohol consumption. Further, approximately 74% of employees resided in Utah, which has unique characteristics that may be associated with the variables considered in this study (e.g., high altitude living and low tobacco use). In addition, we were unable to evaluate people 65 years or older because as they retired, they tended to opt out of the DMBA insurance program.

The number of employees who filed healthcare claims for speech disturbances and voice disorders was small. Assessing relationships between these cases and certain rare mental health conditions were limited by small numbers. Further, rates of upper respiratory related symptoms and mental illness and sleep disorders may be biased downward because less severe cases may not have sought medical care and, thus, not be represented in the results.

Not all confounders of the relationships assessed in this study are known or were considered. Only the more prominent confounders identified in other studies were included. The database did not contain information about lifestyle factors and social determinants of health, so they were not included as potential confounders.

Finally, the study design limited us to identifying statistical associations and not causal relationships.”

Round 2

Reviewer 1 Report

Comments and Suggestions for Authors

Thanks for responding to the comment. Please carry forward the exploration of the topic. All the best.

Reviewer 2 Report

Comments and Suggestions for Authors

Thank you for incorporating the annotations. I consider your article is now clearer.

Reviewer 4 Report

Comments and Suggestions for Authors

Dear authors, excellent proofreading work. You need to correct footnote #45 by adapting it to the writing style of all the other notes (remove the doi code indication). Otherwise, for me, it is immediately publishable.